# Inferring the extinction risk of marine fish to inform global conservation priorities

**Nicolas Loiseau** [1]*, **David Mouillot**[1,2], **Laure Velez**[1], **Raphaël Seguin**[1], **Nicolas Casajus**[3], **Camille Coux**[3], **Camille Albouy**[4,5], **Thomas Claverie**[1,6], **Agnès Duhamet**[1,7], **Valentine Fleure**[1,8], **Juliette Langlois**[1], **Sébastien Villéger**[1], **Nicolas Mouquet**[1,3]

**1** MARBEC, Univ Montpellier, CNRS, Ifremer, IRD, Montpellier, France, **2** Institut Universitaire de France, Paris, France, **3** FRB–CESAB, Montpellier, France, **4** Ecosystems and Landscape Evolution, Department of Environmental Systems Science, Institute of Terrestrial Ecosystems, ETH Zürich, Zürich, Switzerland, **5** Unit of Land Change Science, Swiss Federal Research Institute WSL, Birmensdorf, Switzerland, **6** ENTROPIE, Univ La Réunion, IRD, IFREMER, Univ Nouvelle-Calédonie, CNRS, Saint-Denis, France CUFR of Mayotte, Dembeni, France, **7** CEFE, Univ Montpellier, CNRS, EPHE-PSL University, IRD, Montpellier, France, **8** ZooParc de Beauval & Beauval Nature, Saint-Aignan, France

* nicolas.loiseau@cnrs.fr

## Abstract

While extinction risk categorization is fundamental for building robust conservation planning for marine fishes, empirical data on occurrence and vulnerability to disturbances are still lacking for most marine teleost fish species, preventing the assessment of their International Union for the Conservation of Nature (IUCN) status. In this article, we predicted the IUCN status of marine fishes based on two machine learning algorithms, trained with available species occurrences, biological traits, taxonomy, and human uses. We found that extinction risk for marine fish species is higher than initially estimated by the IUCN, increasing from 2.5% to 12.7%. Species predicted as Threatened were mainly characterized by a small geographic range, a relatively large body size, and a low growth rate. Hotspots of predicted Threatened species peaked mainly in the South China Sea, the Philippine Sea, the Celebes Sea, the west coast Australia and North America. We also explored the consequences of including these predicted species' IUCN status in the prioritization of marine protected areas through conservation planning. We found a marked increase in prioritization ranks for subpolar and polar regions despite their low species richness. We suggest to integrate multifactorial ensemble learning to assess species extinction risk and offer a more complete view of endangered taxonomic groups to ultimately reach global conservation targets like the extending coverage of protected areas where species are the most vulnerable.

## Introduction

Target 3 of the Convention on Biological Diversity's Kunming-Montreal Global Biodiversity Framework—adopted in December 2022—aims to increase the global coverage of protected areas (PAs) to at least 30% by 2030 (hereafter referred to as 30 × 30), with the ultimate goal to

**Data Availability Statement:** All data, code, figures are available from github https://github.com/LoiseauN/FISHUCN_clean The data and code are

also available from https://zenodo.org/records/12783687.

**Funding:** NL is supported by the Fondation pour la Recherche sur la Biodiversité (FRB) and La region Occitanie (BiodivOc) in the context of the CESAB project 'Creating a global database of fish functional traits: integrating physiology and ecology across aquatic ecosystems' (PHENOFISH). DM was partly funded through the 2017–2018 Belmont Forum and BiodivERsA REEF-FUTURES project under the BiodivScen ERA-Net COFUND programme and with funding from ANR, DFG, NSF, Royal Society, ERC and NSERC. The funders had no role in study design, data collection and analysis, decision to publish, or preparation of the manuscript.

**Competing interests:** The authors have declared that no competing interests exist.

**Abbreviations:** ANN, artificial neural network; CAZ, core-area zonation; CBD, Convention on Biological Diversity's; DD, data deficient; IUCN, International Union for the Conservation of Nature; GCM, global circulation model; MPA, marine protected area; NE, not been evaluated; PA, protected area; RF, random forest; SDM, species distribution model.

deliver benefits for nature and people where the needs are the most pressing. Consequently, prioritizing the establishment of new PAs and the strategic use of limited conservation resources are crucial to mitigate the ongoing global biodiversity crisis [1,2]. However, a strategy that protects as many species as possible—regardless of the risk of extinction—may lead to a different prioritization of new protected areas than a strategy that emphasizes the protection of the most threatened species. To address this issue, assessing species extinction risk is of primary importance despite persistent challenges [3,4]. In this regard, the International Union for the Conservation of Nature (IUCN) regularly updates the global Red List (www.iucnredlist.org), which classifies species by their increasing extinction risk (Vulnerable, Endangered, and Critically Endangered) mainly based on their population and geographic range size. Yet, this classification requires extensive knowledge and many species with limited information are considered as Data Deficient.

In 2023, the IUCN Red List contains 150,388 species (including mammals, birds, reptiles, amphibians, fishes, insects, and plants) classified in 3 categories: (1) "Threatened," which encompasses the Critically Endangered, Endangered, and Vulnerable IUCN categories; (2) "Non-Threatened," which includes the Least Concern and Near Threatened IUCN categories; and (3) Data Deficient (DD), which contains the largest number of species. In addition, most animal biodiversity (estimated to >1.8 million species of metazoans) has not been evaluated (NE). Even for the most studied vertebrate taxa such as mammals and reptiles, the proportion of DD or NE species, hereafter grouped under the DDNE category, is still high (respectively 22.9% and 27.8%, Fig 1). This knowledge gap may leave threatened species out of conservation priorities and bias conservation prioritization based on extinction risk. For example, in 2014 nearly half (454 species) of sharks and rays (chondrichthyan) were still NE, and after a new assessment of this group in 2021, 37.5% (against 17%) were classified as threatened by extinction [5,6]. Given the global biodiversity crisis, the ever-increasing number of threatened species urges for a greater collective effort to fill these IUCN classification gaps to better guide conservation planning. Yet, this ambitious goal is far from being reachable given the millions of species on Earth and the inherent difficulty to obtain accurate information for most of them due to their remote or hardly accessible habitat (e.g., deep sea, high mountain), behavior (e.g., elusive, nocturnal), body size (e.g., <1 cm), or rarity (e.g., endemic). Alternative methods are thus needed to predict species extinction risk status and to ultimately fuel global conservation prioritization algorithms [7–10].

Some methods have been proposed to infer the IUCN status of unassessed (DD and NE) species [11,5]. Species distribution models (SDMs) can predict the spatial distribution or the temporal dynamics of populations according to their environmental or ecological niches [12]. However, these models require a large spectrum of variables (such as climatic niche, habitat, human footprint index, geographic distribution, phylogeny, and traits) to eventually predict the loss of suitable habitat and then extinction risk [13]. Modeling the distribution of rare, and therefore the most threatened species [14,15] is also associated with a high level of uncertainty due to the low number of occurrences, which can lead to model overfitting and inaccuracy [16]. SDMs also require climate projections from global circulation models (GCMs) that in turn rely on socioeconomic scenarios (RCPs) making the overall process challenging to achieve for many taxonomic groups [17]. Meanwhile, the prediction of species IUCN status has benefited from the development of machine learning models with the underlying assumption that unassessed species are likely to share a similar IUCN status to those having similar biological traits, geographic distribution, or evolutionary history [18]. For example, machine learning models could predict IUCN categories for mammals, amphibians, reptiles [19–22], sharks, rays [23], and orchids [18] with high accuracy (e.g., up to 92% for terrestrial mammals [19]).

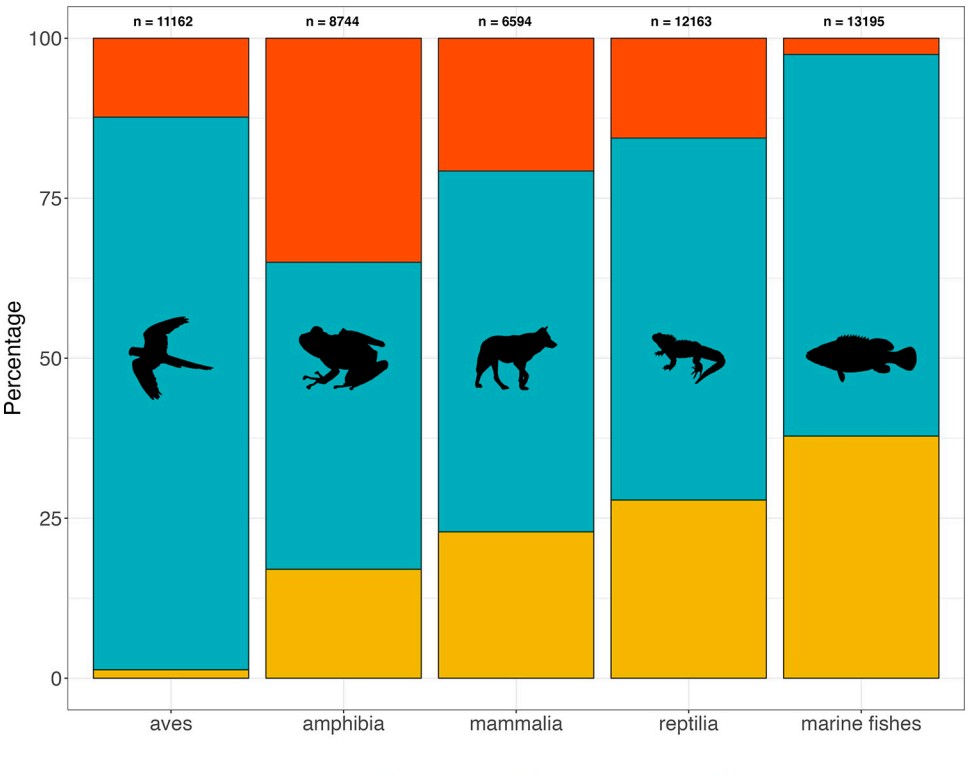

**Fig 1. IUCN status (IUCN 2024) among birds, reptiles, mammals, amphibians, and marine fishes.** IUCN-assessed species were classified into 3 categories: "Threatened" gathering the Critically Endangered, Endangered, and Vulnerable IUCN categories; "Non-Threatened" gathering the Least Concern and Near Threatened IUCN categories; and "DDNE" gathering the Data Deficient and Not Evaluated IUCN categories. Because some species were not present in the IUCN Red List, we updated the species list with https://www.birdlife.org/, http://www.reptile-database.org/db-info/SpeciesStat.html, https://www.mammaldiversity.org/, https://amphibiansoftheworld.amnh.org/. Icons were generated using R (rphylopic package) and are under the Creative Commons Attribution 4.0 International (CC BY 4.0) License. The data underlying this figure can be found in https://zenodo.org/records/12783687. IUCN, International Union for the Conservation of Nature.

This IUCN categorization is overdue and urgently needed for marine fishes which are highly diverse ($N > 15,000$ species), among which many are facing multiple threats [24–26] and support key contributions to nature and people like nutrient cycling, carbon sequestration, ecosystem resilience, productivity, as well as nutritional and cultural values [27–29]. Among vertebrates, marine teleost fishes have the highest proportion of DDNE species (38%, $n = 4,992$ of 13,195, Fig 1). Ultimately, a more extensive and accurate IUCN classification of marine fishes could reevaluate the prioritization of new protected areas. This is particularly relevant concerning the new agenda to protect 30% of marine waters before 2030 ($30 \times 30$ target, Conference of the Parties to the Convention on Biological Diversity, COP 15).

Here, we used a combination of random forest model (RF) and artificial neural network algorithm (ANN) to predict extinction risk of 4,992 DDNE marine fish species (Fig 2) based on their occurrence data, traits (i.e., body size, trophic position), taxonomy, and human uses. We then addressed 4 principal questions for the conservation of marine fishes: 1. Which attributes of a species are the best predictors of their extinction risk? 2. How does the addition of species predicted as Threatened (Critically Endangered, Endangered, and Vulnerable) change the distribution of hotspots of extinction risk? 3. Does the current network of marine protected

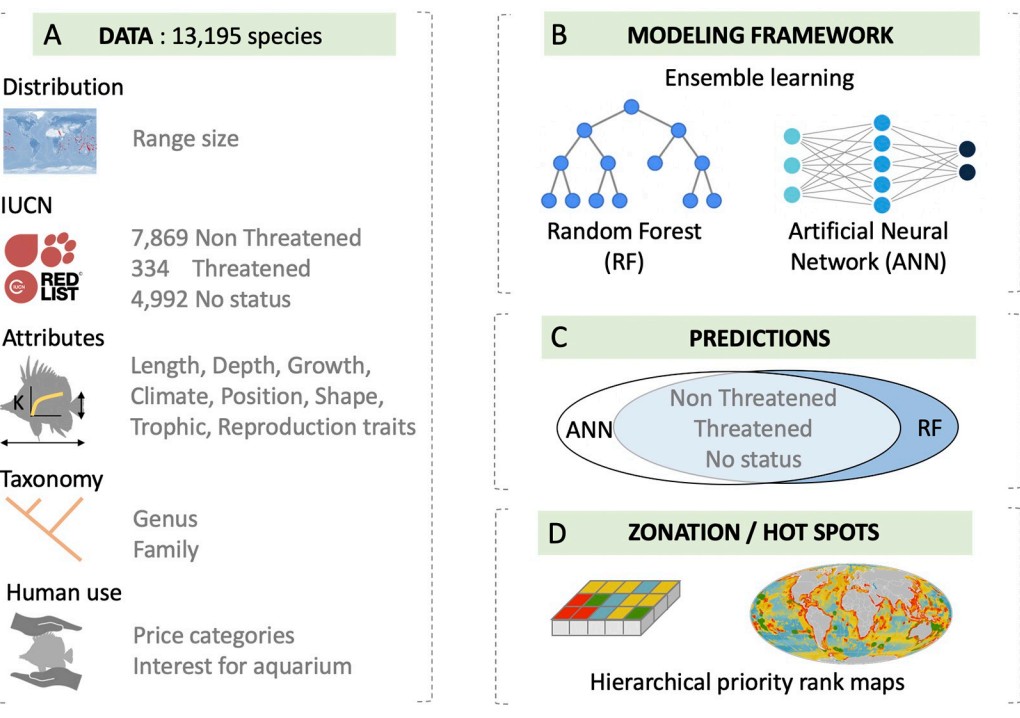

**Fig 2. Illustration of our modeling framework to infer the IUCN status of 4,992 Data Deficient and Not Evaluated marine fishes.** Using available occurrence data, species biological traits, taxonomy, and human uses (A), we built an ensemble learning model using RF and ANN (B) to predict the IUCN status of marine fishes using complementary decisions between ANN and RF outputs (C). Then, we explored the consequences of including the predicted threatened species on the areas currently prioritized by conservation planning (D). See methods for a complete description of these steps. Map was created using R package rnaturalearth (https://www.naturalearthdata.com/). ANN, artificial neural network; IUCN, International Union for the Conservation of Nature; RF, random forest.

areas (MPAs) cover this threatened marine fish diversity? 4. To which extent does the classification of marine fishes with no IUCN status modify conservation priorities to meet the 30 × 30 target?

# Results

## Predicting the IUCN status

For both RF and ANN, we performed cross-validation, it order to determine their accuracy (to predict IUCN status) based on the proportion of false positives (proportion of Non-Threatened predicted as Threatened) and false negatives (proportion of Threatened predicted as Non-Threatened). We found that RF models better predicted species extinction risk (accuracy of 0.77) than ANN algorithms (accuracy of 0.70). RF achieved a high rate of true-positive predictions (77.3%, SD = 3.89%, Figs A and B in S1 Text) in cross-validation tests and a low rate of false-positives (12%, SD = 3.02%, Figs A and B in S1 Text) and false-negatives (10.7%, SD = 3.1, Figs A and B in S1 Text). ANN achieved a lower rate of true-positives (70%, SD = 3.67%, Figs A and B in S1 Text) with a higher rate of false-positives (13.8%, SD = 3.62%, Figs A and B in S1 Text) and false-negatives (16.2%, SD = 3.51%, Figs A and B in S1 Text). RF models were able to predict the IUCN status of fewer species (2,324 Non-Threatened and 1,440 Threatened species) than ANN algorithms (2,677 Non-Threatened and 1,294 Threatened species).

# BEFORE          AFTER

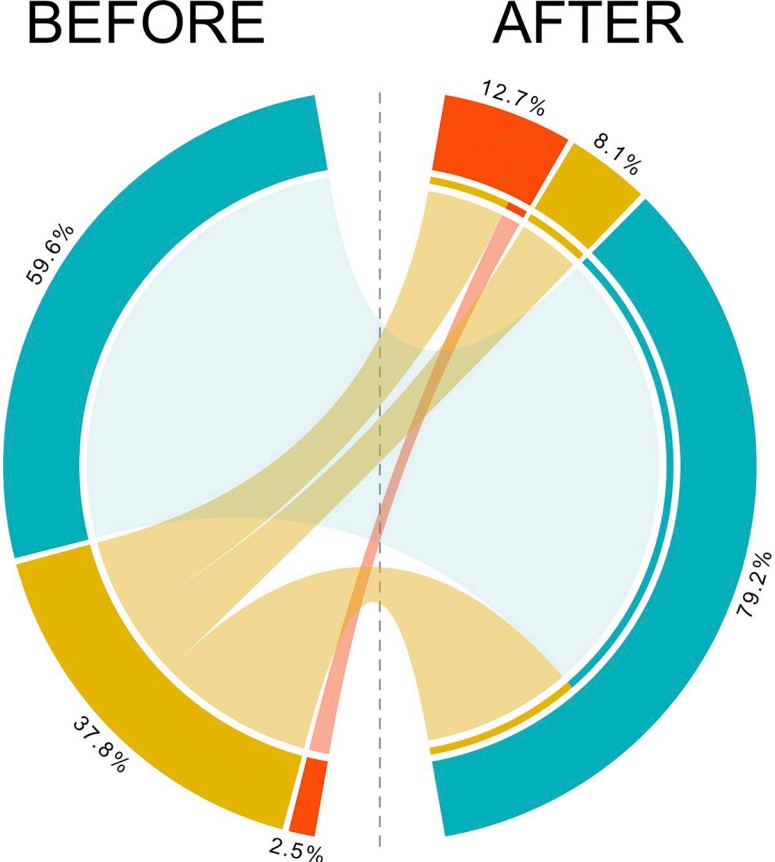

**Fig 3.** Distribution of species among the extinction risk IUCN categories before (i.e., based on current Red List) and after predictions. Species are grouped into 3 broad categories following the IUCN status: "Threatened" (red) including Critically Endangered, Endangered, and Vulnerable species; "Non-Threatened" (blue) including Least Concern and Near Threatened species; DDNE (gold) merging Data Deficient and Not Evaluated species. The DDNE category after prediction refers to the species for which the 2 algorithms disagreed, no prediction could be made. The data underlying this figure can be found in https://zenodo.org/records/12783687. IUCN, International Union for the Conservation of Nature.

After the within-consensus framework (i.e., cross-check within each algorithm, see Methods), we combined ANN and RF outputs using a complementary decision tree: A status (Threatened or Non-Threatened) was attributed to a given species when both methods predicted the same status or when only one method was able to accurately predict a status while the DDNE status was kept when the predictions of the two algorithms differed (S1 Table). Predictions differed for 573 species that remained DDNE. Overall, out of the 4,640 DDNE species (4,992 minus the 352 unpredictable species with too many missing trait values and that remained DDNE, see Methods), 1,337 were categorized as Threatened and 2,582 as Non-Threatened, resulting in a much higher proportion of threatened species than expected from the current IUCN categorization (Fig 3). Overall, the number of DDNE species was reduced by 78.5% (1,073 out of 4,992 species remained DDNE), the number of Threatened species increased by 400% (from 334 to 1,671) while the number of Non-Threatened species increased only by 34.8% (from 7,750 to 10,451). We also applied a consensus decision from ANN and RF outputs and found that even if the number of predicted species decreased, the number of Threatened species increased (824) disproportionately compared to the number of Non-Threatened species (1,846 see Fig C in S1 Text).

## Which species attributes predict the IUCN status?

RF models provided information about which features were the best to predict species IUCN status (Fig 4). Species predicted as Threatened were mainly characterized by a small geographic range, a relatively large body size, and a low growth rate (Fig 4). The likelihood of species being Threatened increased with their preference for very shallow habitats (Fig 4). We also found that the Family variable contributed to the prediction of the IUCN status probably due to the strong phylogenetic conservatism in size, growth rate, and vertical position. Closely phylogenetically related species were indeed significantly more likely to share the same IUCN status than distantly related species (Fig D in S1 Text). Some families gathered a high proportion of species predicted as Threatened (phylogenetic signal D index = 0.68 ± 0.01). For example, we predicted 19 species of Bythitidae out of 28 (67.8%) and 56 species of Serranidae out of 131 (42.7%), as Threatened, thus changing their previous DDNE classification (Fig E in S1 Text). Some cryptobenthic fish families like Gobiidae, Gobiesocidae, Blennidae had also an important proportion of species predicted as Threatened. Conversely, some families hosted a low proportion of species predicted as Threatened. For example, none of the 17 DDNE Myctophidae were predicted as Threatened. Overall, we found that species predicted as Non-Threatened or remaining DDNE were also clustered across the phylogeny ($D_{NonT}$ index 0.59 ± 0.01; $D_{DDNE}$ 0.90 ± 0.01, Fig D in S1 Text).

## Where are the hotspots of fish species extinction risk?

Before gap-filling prediction, Threatened species were mainly aggregated in the Caribbean, the South China Sea, the Philippine Sea and the Celebes Sea. After prediction, new hotspots of Threatened species emerged in western Australia and on the west coast of North America (Fig 5A). Overall, the China Sea, Philippine Sea, and south Japan aggregated the highest number of species predicted as Threatened (Fig 5A). The distribution of Non-Threatened species before

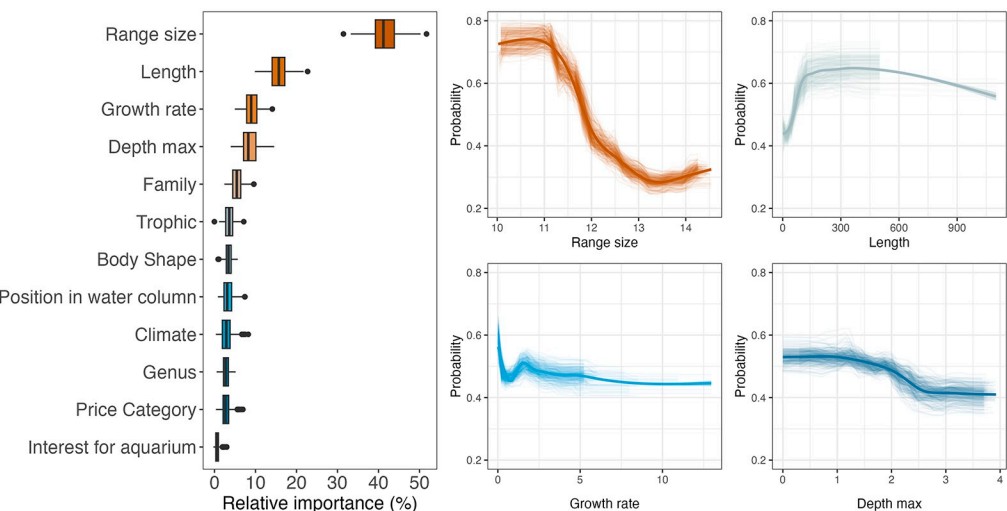

**Fig 4. Key species attributes predicting the IUCN status of marine fishes.** Relative importance (in %) of 12 biological traits and human uses in the 240 random forest models (left) and partial plots showing the influence of the 4 main attributes on the IUCN status of marine fishes (here the probability to be Threatened on the Y-axis). The data underlying this figure can be found in https://zenodo.org/records/12783687. IUCN, International Union for the Conservation of Nature.

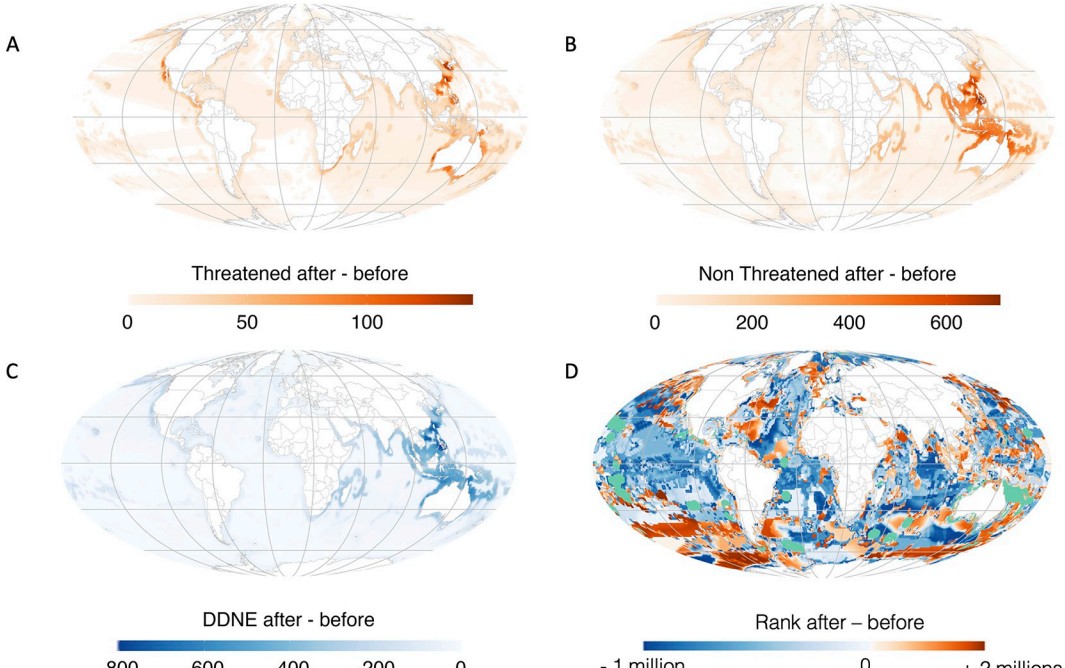

**Fig 5.** Spatial distribution of the difference in the number of Threatened (A), Non-Threatened (B), and DDNE (C) species before and after prediction, and in the prioritization rank after the prediction (rank after—rank before, D). The color gradient indicates the difference in the number of species or prioritization ranking from red (value of the cell is higher after prediction) to blue (value of the cell is lower after prediction). Green indicates already protected cells. Maps were created using the R package rnaturalearth (https://www.naturalearthdata.com/). The data underlying this figure can be found in https://zenodo.org/records/12783687.

and after prediction followed the gradient of marine fish richness, with Non-Threatened species peaking in the Indian Ocean and Coral Triangle (Fig 5B). Finally, the remaining DDNE species were mainly aggregated in the China Sea, the Philippine Sea, and in southern Japan (Fig 5C).

Furthermore, we assessed species-specific coverage by the global network of MPAs and target achievement, defined as the proportion of a species' geographic range covered by these protected areas, before and after predictions of species IUCN status. These specific targets were related to species range sizes with the most restricted species needing more coverage (e.g., 100%) than widespread one (e.g., 10%) to avoid extinction. First, we found that regardless of gap-filling status predictions, target achievement, and MPA coverage of Threatened and DDNE species were significantly smaller than for Non-Threatened species (Fig 6, Kruskal–Wallis chi-squared = 617, df = 2, $P < 0.001$). Threatened species were significantly less protected than DDNE species, but to a lower extent (Fig 6, Kruskal–Wallis multiple comparison, Z = 2.87, $P < 0.05$). Second, we observed a decrease in the attainment of target protection for DDNE (Wilcoxon test, W = 2785657, $P < 0.05$) and Threatened species, albeit not significantly (Wilcoxon test, W = 271429, $P = 0.4$), indicating that these species were not as protected as they should be. Average target achievement for Threatened and DDNE species were both equal to 3.6% before IUCN status predictions compared to 2.3% and 2.5%, respectively after predictions. However, we did not observe a significant difference in the percentage of species-range covered by MPAs for Threatened species before and after IUCN status predictions (Fig 6, Wilcoxon test, W = 271429, $P = 0.4$).

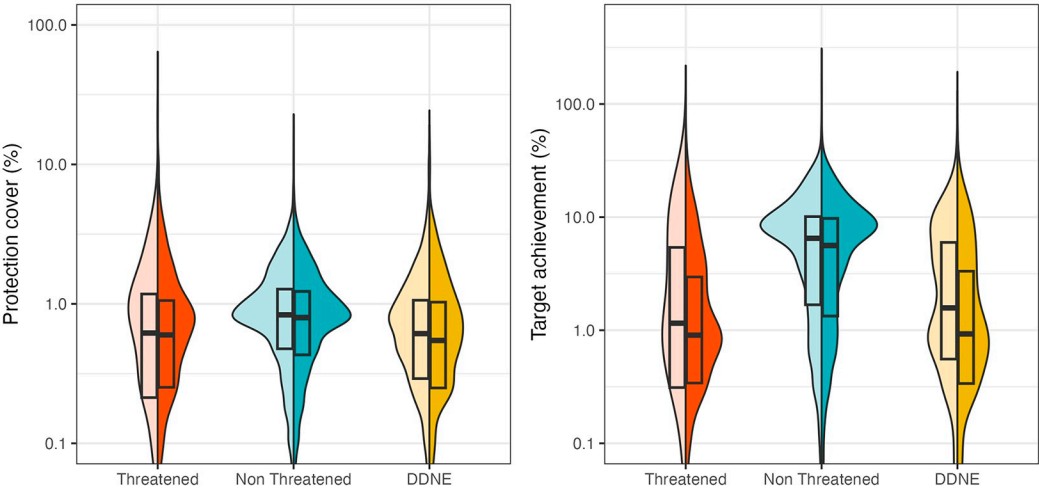

**Fig 6.** Protection status of Threatened, Non-Threatened, and DDNE (data deficient or not evaluated) species before (light colors) and after (dark colors) predictions. (A) Percentage of species protection coverage (proportion of geographical range currently covered by protected areas), and (B) species target achievement (extent to which species are represented within protected areas regarding their restrictiveness). The data underlying this figure can be found in https://zenodo.org/records/12783687.

### Influence of IUCN status predictions on global conservation planning

To test whether the predictions of the IUCN status for currently DDNE species could disrupt conservation planning, we compared conservation priorities based on assessed species IUCN status (Scenario 1: 7,869 Non-Threatened, 4,992 DDNE, and 334 Threatened species) with conservation priorities accounting for new species predicted IUCN status (Scenario 2: 10,451 Non-Threatened, 1,073 DDNE, and 1,671 Threatened species). We used the Zonation algorithm that identifies which locations in a seascape are most important for protecting threatened biodiversity (see Methods). Zonation ranks locations (hereafter "cells") in function of their importance for conservation. The least valuable cells received the lowest ranks (0), and those having the highest priority reached the highest ranks. We fixed species priority weights to 1 for Non-Threatened species, 6 for Threatened species, and 2 for DDNE species because there is accumulating evidence that DD and also NE species are more at risk than Non-Threatened [30] (see Methods for details on the weighting and Fig G in S1 Text). We then compared the ranking of each cell between both scenarios (with and without predicted IUCN status).

Overall, we found a marked change in conservation priority ranking after species IUCN status predictions. This is particularly true at low and intermediate values of species richness where the ranking is more likely to shift (Fig 7A and Fig H in S1 Text). By plotting the delta ranks (rank after minus rank before) on the latitudinal gradient, we found that the major changes in high ranking were at low (<30°) and high latitudes (>50°) corresponding to temperate and polar climatic zones for which species richness is the lowest, as well as in Pacific islands (Figs 5D, 7B, and Fig H in S1 Text).

### Discussion

Models will never replace a direct evaluation of species extinction risk based on empirical robust data, but coupling machine and deep learning methods offer a unique opportunity to provide a rapid, extensive, and cost-effective evaluation of extinction status [20] while also

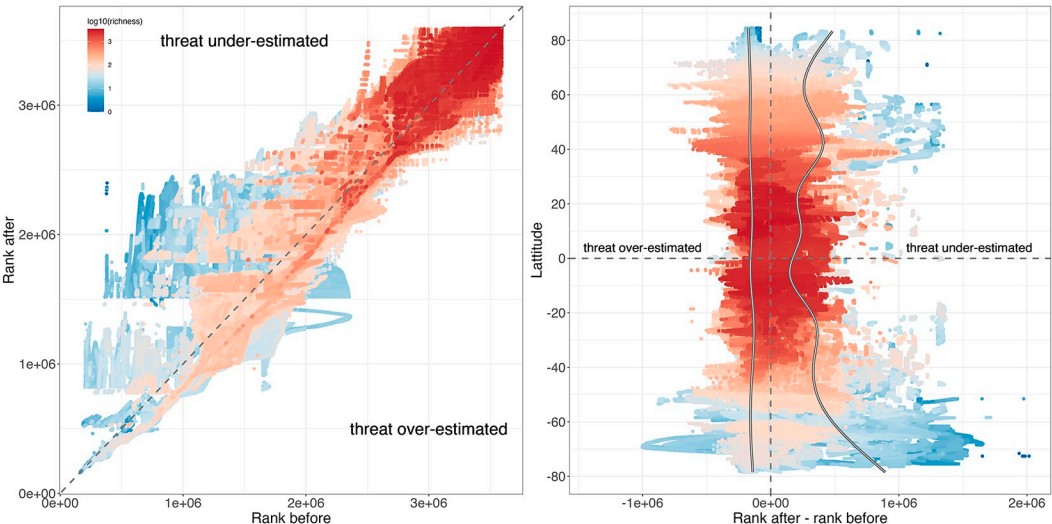

**Fig 7. Change in global Zonation priority ranking for the 3,594,495 marine cells (10 km/10 km), after predictions of marine fish IUCN status.** (**A**) Relationship between ranks before and after predictions; dots color gradient indicates cell species richness (log10). The dashed line represents x = y (i.e., cells above this line have seen their priority rank increasing). (**B**) Relationship between the change in rank (rank after minus rank before) of each cell and its latitude; dots color gradient indicates cell species richness (log10). Lines show for both, negative and positive delta rank values, the relationships with latitude were obtained with a generalized additive model. The data underlying this figure can be found in https://zenodo.org/records/12783687. IUCN, International Union for the Conservation of Nature.

pointing out the species on which data collection and conservation efforts should be prioritized. Several studies have already proposed automated methods to conduct a preliminary assessment of species conservation status based on their attributes or remotely sensed predictors [3,11,18–23,31]. However, to our knowledge, they have not yet been incorporated in the official Red List assessment [4]. We believe that ensemble learning is relevant since it is accurate and conservative. The performance of machine learning algorithms is known to vary based on factors such as the dimensionality of the data set [31]. To address this variability, we suggest a multi-model strategy combining distinct algorithms to leverage their strengths and mitigate their weaknesses. With relatively small data sets, Random Forests can achieve a high level of accuracy, whereas Neural Networks typically require more data to reach a similar level of performance [32]. Conversely, Random Forests show minimal performance improvement beyond a certain data threshold, while Neural Networks generally benefit from larger data sets and continuously improve their accuracy [32]. Random Forests are also advantageous in terms of interpretability, because they highlight which features are the best to predict species IUCN status.

The accuracy of our models (0.77, 0.70 for the RF and ANN, respectively) was slightly lower compared to the binary classifier developed by Borgelt and colleagues [33] for amphibians (85%) or the IUC-NN classifier developed by Zizka and colleagues [18] for orchids (84%). This lower accuracy can be attributed to the limited number of Threatened species included in our training data set whereas, for example, Zizka and colleagues [18] had a significantly larger representation of Threatened species, accounting for 49.7% of their data set. Our data set covered almost all marine fishes, with a very high initial number of Non-Threatened species (7,869) and a relatively low number of Threatened species (334). Furthermore, both Borgelt and colleagues [33] and Zizka and colleagues [18] classifiers were trained using habitat data, which unfortunately were not available for the majority of marine fish species in our study.

Adding information about marine fish habitats should be a priority to increase the accuracy of our models which can be offered by the recent developments in satellite or acoustic imagery [34]. The robustness of our predictions was increased by our decision tree relying on (1) the within-consensus framework (cross-check within each algorithm); and (2) the use of 2 models rather than 1, even if this tends to predict fewer species status than using only 1 model. Indeed, if the predictions between the 2 models differ, we do not provide a status—this applies to 573 species (12% of the 4,640 DDNE in the models). Altogether, the multi-model strategy we proposed appears like a good compromise between accuracy and conservatism to predict IUCN status and should be tested on more taxa to validate its utility as a companion tool for IUCN assessment.

Using only 3 categories—Threatened, Non-Threatened, and DDNE—comes with some limitations. For example, assigning the same weight to Vulnerable or Critically Endangered species despite their distinct status may not accurately reflect the varying levels of conservation effort required for their protection. This was again linked to the low number of species within these 2 categories (224 VU, 77 EN, and 33 CR) in the original data set. It suggests prioritizing the direct evaluation of species for which we have predicted Threatened status to be able to refine our predictive model. The 573 species for which we could not reach a consensus on both models should be also prioritized for future evaluation since one of the algorithms predicted them as Threatened. Another limitation of our approach stands from having predicted missing species traits with a Random Forest algorithm, which may ultimately lead to a misclassification of some species. However, only traits with a missForest performance exceeding 0.6 ($R$ squared $> 0.6$ for regression or 60% for classification) were attributed, thus minimizing the probability of errors in trait inference. Moreover, we used coarse biological traits already available on Fishbase. Although we acknowledge that extinction probabilities are related to species responses to climate change [26], accounting for such effects would require gathering more ecophysiological-based traits [25,26] (i.e., metabolic rates, thermal optimum, reproduction). While these traits were not available for most marine species, the growing availability of fish traits will ultimately make categorical predictions of conservation status more effective in the near future. Finally, we used species range maps provided by Albouy and colleagues [35] which do not perfectly reflect the current distribution of marine fishes but are nevertheless based on a robust method to minimize errors in the original OBIS data set (see Methods). Since OBIS is continuously aggregating new observations, it cannot assess range contractions or regional extirpations due to environmental shifts or overexploitation [36,37], which could result in an underestimation of the number of Threatened species.

Altogether, the limitations of the in silico species-risk assessment open opportunities for improvements and inputs from the organization (IUCN) for which the predictions are made, which could trigger a positive virtuous loop and lead to an effective in situ/silico assessment of species extinction risks. Indeed, our prediction of the IUCN status for marine fish species shows a fivefold increase of fish species with a Threatened status (from 334 to 1,671 Threatened species), so from 2.5% to 12.7% of total species richness. Meanwhile, the number of species with Non-Threatened (NT) status only increased by 34.8% (from 7,869 to 10,451 Non-Threatened species). Overall, 1,073 species remained DDNE (8.1%), which suggests that there is still some potential to increase the accuracy of our predictive model. Even when we applied a consensus decision to the ANN and RF outputs, we found that the number of Threatened species increased disproportionately, from 2.5% to 8.8%.

Given the strong phylogenetic conservatism of environmental and trophic niches among marine fishes [38], we expected that the assessment of a species as Threatened would often coincide with the status of its closest relatives. Thus, a strong proportion of species from some families like Sebastidae, Bythitidae, or Serranidae (Fig C in S1 Text) has been predicted as

Threatened. Additionally, some cryptobenthic fishes families like Gobiidae, Gobiesocidae, Blennidae also host an important number of species predicted as Threatened (S3 Fig C in S1 Text). Cryptobenthic fishes fulfill crucial ecological roles, particularly in the dynamics of trophic interactions and the overall functioning of reef ecosystems [39]. Due to their elusive behavior and reliance on specific habitats that restrict the assessment of their populations, certain species within these families may be undergoing a silent extinction process, underscoring the urgent need for increased evaluation efforts on these species.

Although we found that closely related species were more similar in their IUCN categories than distantly related species (for Threatened, Non-Threatened, and DDNE), we found that taxonomy (family and genus) was not the best predictor of IUCN categories. Rather, species attributes (being relatively common in vulnerability assessment), geographical range, body size [4], and growth rate [40] were much better predictors. Note that species traits indirectly include phylogenetic information, which might reduce the importance of taxonomy in our models. This result highlights which ecological species attributes should be assessed in priority to enhance our ability to accurately predict and detect threatened species. It can also determine which species should be assessed in priority by IUCN experts as a precautionary principle: fishes with small geographical range (already well used as a criterion in the IUCN assessments), large body size, and slow growth rate (known to be correlated [36]).

By mapping the distribution of the predicted species, we provide 2 crucial pieces of information for future evaluation: the hotspots of predicted Threatened species where conservation effort should increase, and the hotspots of DDNE species where research effort should increase. After IUCN status predictions, Threatened species predominantly occurred in the tropics, peaking in Indonesian islands, West-Australia, and in the China Sea, as well as in the west coast of America. For these regions, the establishment or reinforcement of effective marine protected areas should be prioritized, along with increased research effort. Conversely, the gain in Threatened species after the prediction was lower in the Caribbean Sea. This could be explained by a higher research effort [41] in this part of the world leading to better classification of IUCN fish status. Despite being recognized worldwide as a hotspot of diversity, we found that the coral triangle was also a hotspot of DDNE after IUCN status predictions. Since the most important changes in sea surface temperature are occurring in this part of the world [42], the risk of species extinction here is particularly high and the status of these remaining DDNE species should be prioritized. The China Sea also requires a particular effort to provide new information on species to assess their extinction risk.

Because the IUCN Red List is an instrument for conservation planning, management, monitoring, and decision-making [7], we expected that target achievement would be higher for Threatened species. Meanwhile, DDNE species are typically overlooked in conservation planning [19], with the implicit assumption that extinction risk for DDNE and Non-Threatened species is similar [43]. By reducing the number of DDNE species and increasing the number of Threatened species, we show that Threatened marine fish species generally reach low conservation target achievement and are poorly covered by current protected areas. This strongly contrasts with the higher level of target achievement observed for Non-Threatened species (Fig 5B).

We also examined the extent to which inclusion of predicted Threatened species affected the spatial prioritization to conserve worldwide marine fish diversity. Since the prioritization algorithm is strongly influenced by the number of species [44], we found that the ranking of the richest regions was marginally modified. However, we found that low- to middle-rank regions were increasing in conservation priority, revealing the importance of protecting subpolar, polar, and Pacific Island areas as well (Fig 7). Specifically, a strong shift in conservation priority was observed in the subpolar and polar regions of the Southern Hemisphere. Since the Antarctic region is typically not subjected to many global agreements (such as the Convention

on Biological Diversity's (CBD) Aichi Targets of the Strategic Plan for Biodiversity 2011–2020), our results advocate for a deeper evaluation of the conservation status of marine fish species in this region. The strong velocity of isotherm and species range shifts due to climate change observed in these cold waters [45] also poses a significant challenge to the success of ambitious conservation strategies. Some strong changes in prioritization were also observed close to the Pacific Islands. Given that only 13% of marine island areas are currently designated as protected, and that half of all islands lack any protected areas [46], it is likely that fish species in these areas face even greater threats than what our framework predicts. This highlights the urgent need for a significant risk assessment by the IUCN of fish fauna occurring close to islands.

IUCN will increase its efforts in the next decade to complete the extinction risk assessment for many taxa, but there will still be millions of other species to assess, which is simply not feasible given the IUCN standards. Also, paradoxically, the highly publicized annual update of the IUCN Red List brings to the public biased information on the state of biodiversity with a much greater emphasis on few taxa, such as vertebrates [47]. Consequently, whatever efforts the IUCN puts into assessing species from other taxa and communicating about the inherent biases of the Red List, it is now essential to develop a pragmatic approach to extend extinction risk assessments towards overlooked taxa. This means bringing some in silico assessments into the IUCN procedure. As illustrated here, combining large-scale data sets into a multi machine learning framework allows to at once provide reliable extinction risk status for species not evaluated by the IUCN, and point out which species attributes and geographic regions should be assessed in priority to increase the accuracy of the modeling approach and predict status for still unpredictable species. Understanding all the steps associated with this in silico assessment of extinction threats for many different taxa (see for example, Borgelt and colleagues) [33] will also provide a more comprehensive understanding of species conservation status [4]. Such an integrated strategy will improve prioritizing efforts as well as allocating resources effectively to mitigate extinction risk globally [4]. We also advocate for the IUCN to integrate recent developments in forecasting species extinction risks (including our approach) into a synthetic new index of "predicted IUCN status" that could complement the actual "measured IUCN status." This change would help provide the scientific community with more data on species extinction risks. In addition, governments and the broader public would have their attention brought to a more balanced taxa perception of the ongoing biodiversity crisis.

## Material and methods

### Occurrences and species ranges

We used the data from Albouy and colleagues [35] which were sourced from OBIS (http://www.iobis.org) on August 27, 2014. We chose to work with data that is highly accurate, even if it is not the most recent. They collected a total of 16,238,200 occurrence records from 34,883 entries. To ensure data quality, they performed data cleaning procedures that involved identifying and resolving issues such as synonyms, misspellings, and removing rare species (those with only 1 occurrence). This resulted in a set of 11,503,257 occurrences for 11,345 fish species around the world. As the OBIS database did not represent the tropical assemblage of fish well enough, they merged it with the Gaspar database that encompasses 6,316 coral reef species [48]. Additionally, we limited our analysis to species known to inhabit marine environments based on FishBase [49]. As a result, we obtained a data set representing 14,035 fish species from around the world. In this pool of species, we still found 840 freshwater and brackish water species. We removed these species and worked on a pool of 13,195 marine fish teleost species.

To counteract certain known biases in OBIS data (for example, not all species/regions are equally represented), we reconstructed distribution maps for each species, defined as the

convex polygon surrounding the area where each species was observed [35]. The resulting polygon was divided into 4 parts across the world to integrate possible discontinuity between the 2 hemispheres, as well as the Atlantic and Pacific Oceans. We then refined each species distribution map by removing areas where maximal depths fell outside the minimum or maximum known depth range of the species. Finally, we aggregated fish range maps on a 1° grid resolution for the 13,195 marine fish teleost species [35]. We then projected and downscaled all species ranges on a 10 × 10 km resolution grid using the Mollweide projection, which is an equal-area pseudocylindrical projection. We also used this grid to compute the range sizes of each species. The minimum range for a species was 14,900 km [2] (i.e., 149 cells).

## Conservation status

We used the *rRedList* (*v0.7.0*) R package to obtain the updated IUCN status of the 13,195 remaining fishes. The number of fish classified in several IUCN categories was too small to allow us to predict precisely each of these categories. Thus, we grouped species in 3 categories: 1. Critically Endangered, Endangered, and Vulnerable species as "Threatened"; 2. Least Concern and Near Threatened species as "Non-Threatened"; and 3. Data Deficient and Not Evaluated species as "DDNE". In total, 7,869 species were classified as Non-Threatened, 334 as Threatened, and 4,992 were DDNE.

## Species attributes and human uses

We selected 9 species attributes to describe the biology and ecology of species: (1) growth rates (K); (2) the maximum length; (3) the mode of reproduction (dioecism, protandry, protogyny, true hermaphroditism, and parthenogenesis); (4) the maximum and the minimum depth at which species was observed; (5) the reproduction fertilization (refers to where the egg and sperm meet, which may be: external, internal (in the oviduct), in the mouth, in a brood pouch or similar structure, or elsewhere); (6) the body shape; (7) trophic level; (8) climate niche; and (9) position in the water column. We also used information on human uses, specifically price categories (as a proxy of fishing pressure) and interest for aquariums. Finally, the genus and the family of the species were added in models. We extracted all these values (see Table A in S1 Text) from FishBase [49] by using the *rfishbase* (*v4.1.1*) R package.

Because deep learning is not able to handle missing values, we filled out NAs in our 11 predictor variables by applying a Random Forest imputation algorithm (*missForest v1.4* R package). We tested the missForest performance for each predictor variable using a cross validation approach. We ran the missForest on 80% of the complete data (training) and tested its performance on the remaining 20% (testing).

## Phylogeny

We used phylogenetic fish classification from Rabosky and colleagues [50] with updates by Chang and colleagues [51]. Using *fishtree* (*v0.3.4*), we extracted the 100 phylogenetic trees. To measure phylogenetic signal of IUCN status we computed the D index [51], on the 100 phylogenetic trees using the R *phylo.d()* function in the R package *caper* (*v.2.0.6*). The D index equals to 1 if the predicted IUCN status has a phylogenetic random distribution and equals 0 if the predicted IUCN is clumped into the phylogeny.

## Models and predictions

We used ensemble machine learning coupling RF and ANN applied on available occurrence data, species attributes, genus, and family level to predict conservation status. Out of the

13,195 species of marine teleost fishes, 481 (17 Threatened, 119 Non-Threatened, and 352 DDNE) had too many missing values to be incorporated in the predictive framework but were kept for others analyses. Our data set was highly imbalanced; of the remaining 12,714 species, 324 were classified as Threatened, 7,750 as Non-Threatened, and 4,640 remained DDNE. Therefore, we divided the data set into 24 down-sampled data sets with the 324 Threatened species and a different subset of 324 Non-Threatened species. First, we implemented RF. We ran RF with 10-fold cross-validation on each of the down-sampled data sets, resulting in a total of 240 RF models. The accuracy was on average 0.77 (see Figs A and B in S1 Text). Then, we predicted IUCN status for DDNE species for each of the 240 down-sampled data sets. We attributed an IUCN status only to species for which there was a consensus higher than 80% over the 240 RF models. For deep learning, the same framework, features, and data sets as the RF approach were used. We implemented an ANN using the *cito* (v1.1) R package [52]. The accuracy of the ANN was 0.70. We also ran 240 models for the 4,640 DDNE species and attributed an IUCN status only to species for which there was a consensus higher than 80% over the 240 ANN models.

Random Forest and ANN outputs were then used in a three-branch complementary decision tree: (1) For a given species, when both algorithms converged, the given predicted status was assigned to the species; (2) when one of the algorithms was not able to predict status (less than 80% of the models of the given algorithm predicted the same classes) but the other one was able to, the predicted status of the latter one was assigned to the species; (3) when both algorithms diverged, DDNE was assigned to the given species. To test sensitivity of our results, we also applied a consensus approach where for a status to be given (Threatened or Non-Threatened), both machine and deep learning had to predict the same result (when one of the algorithms was not able to predict status, DDNE was assigned, see Fig C in S1 Text).

## Protection and gap analysis

We performed 2 complementary analyses to estimate the extent to which the current marine protected area network covers fish biodiversity. First, we looked at the proportion of geographical range currently covered by protected areas for each species (extracted from the World Database on Protected Areas (WDPA)). We restricted analyses to protected areas classified as Ia, Ib, II, III, IV by IUCN. Second, because species do not require the same conservation effort, we carried out a gap analysis following the methodology proposed in Guilhaumon and colleagues [53]. We defined species-specific conservation targets based on species' range sizes because spatially restricted species require more coverage than widespread species to secure their persistence [53,54]. This species-specific conservation target is expressed as the proportion of a given species' geographical ranges that had to be covered by a protected areas network. Hence, following previous works on gap analysis [53,54], we set conservation targets to be inversely proportional to log-transformed species' range sizes. Following Jones and colleagues [54], we set that species with the range <10,000 km [2] needed 100% of their range to be protected, whereas species with the range >390,000 km [2] only needed 10%. We fitted a linear regression between these 2 values to define a specific target for each species. The proportion of range currently covered by protected areas for each species was divided by the defined target to estimate species target achievement (i.e., percentage of defined targets for each species realized).

## Spatial conservation prioritization

To know how the addition of Threatened predicted species modifies the conservation planning scenarios, we ran the spatial conservation planning Zonation 4.0 [55]. Zonation algorithm

identifies which locations are most important for retaining threatened biodiversity. Specifically, we used the core-area zonation (CAZ) algorithm to identify the best possible expansion of the current protected areas network by ranking the unprotected cells from the $10 \times 10$ km grid in order to provide an optimized global representative coverage for biodiversity conservation. This algorithm maximizes the occurrence of a given feature (in this case, fish species) rather than local richness. For each iterative run, the CAZ algorithm prioritized (i.e., highest value) cells that maximized occurrence of each species. We used as input value the raster of each of the 13,195 marine teleost fish species representing their distribution area on a $10 \times 10$ km resolution grid which represents 3,616,356 cells. For each run, we obtained a map where the value of each cell depends on its order of removal during the prioritization algorithm process. To determine the order, the value of a cell was given by the sum of the distribution fractions of the species present multiplied by their weight. We set to 1,000 the wrap factor parameter which is the number of cells removed at each iteration. These priority values were used to identify locations that contribute most to biodiversity representation (i.e., the unprotected cells with a high conservation gain).

Species were weighted proportionally to their IUCN categories following Montesino-Pouzols and colleagues [30] that assigned the following weights to least concern: 1, near threatened: 2, data deficient: 2, vulnerable: 4, endangered: 6, critically endangered: 8. Accordingly, we fixed the weights to 1 for Non-Threatened, 2 for DDNE, and 6 for Threatened species. The weights attributed to the different IUCN categories are suggestive by definition [44]. To test the robustness of our predictions, we performed a sensitivity analysis that gives more weight to species that have been evaluated by IUCN than by our model. We weighted species as follows: for species evaluated by IUCN, we kept a weight of 1 for Non-Threatened, 2 for DDNE, and 6 for Threatened species. Then, we fixed the weights of predicted species as a function of the average proportion of models (between RF and ANN) $p$, and obtained the final attributed status as follows: Threatened = $2 + p$ (rescaled between 2 and 5), Non-Threatened = $2 - p$ (rescaled between 1 and 2) and DDNE species = 2.

We computed prioritizations following 2 scenarios: before and after IUCN Status predictions. The difference between ranks before and after IUCN prediction was plotted and mapped to emphasize the location that increased or decreased in their conservation priority by new species status.

## Supporting information

**S1 Table. Spreadsheet containing the original and inferred IUCN status of the 13,195 marine fishes.**
(XLSX)

**S1 Text. Supplementary information.** Table A in S1 Text. Table summarizing ecological and human-uses traits utilized to predict IUCN status. Figure A in S1 Text. Boxplot representing the percentages of True predictions (TP), False Positives (FP), and False Negatives (FN) of random forest model (RF) and the artificial neural network algorithm (ANN). The data underlying this figure can be found in https://zenodo.org/records/12783687. Figure B in S1 Text. Boxplot representing performance statistics—Accuracy, F1, recall, precision scores of random forest model (RF) and the artificial neural network algorithm (ANN). The data underlying this figure can be found in https://zenodo.org/records/12783687. Figure C in S1 Text. Chord diagram showing the distribution of species within the different categories before and after prediction when ANN and random forests outputs are used in a consensus way. "Threatened" (red) including Critically Endangered, Endangered, and Vulnerable species; "Non-Threatened" (in blue) including Least Concern and Near Threatened species; "No Status" (DDNE; in yellow) merging Data Deficient and Not Evaluated species. A total of 824 species were

predicted as threatened, 1,846 as non-threatened, and 2,322 species remained DDNE. The data underlying this figure can be found in https://zenodo.org/records/12783687. Figure D in S1 Text. IUCN categories of the 4,992 predicted fishes mapped over the phylogeny. Distribution of the values of phylogenetic signals of ecological rarities (index D) computed on 100 trees are plotted in the center of the tree. The figure represents a single phylogeny from the 100 phylogenies generated (see "Methods"). "Threatened" (red) including Critically Endangered, Endangered, and Vulnerable species; "Non-Threatened" (in blue) including Least Concern and Near Threatened species; "No Status" (DDNE; in yellow) merging Data Deficient and Not Evaluated species and "Non predicted" species in gray. The data underlying this Figure can be found in https://zenodo.org/records/12783687. Figure E in S1 Text. Number of species per family predicted as threatened, non-threatened and that remained no-status. Gray represented species with too many trait missing values and not incorporated in the predictive framework. Species are ordered by the number of threatened species. The data underlying this figure can be found in https://zenodo.org/records/12783687. Figure F in S1 Text. Spatial distribution of the difference in number of Threatened species, Non-Threatened species, and DDNE species before and after prediction (consensus decision tree framework). The data underlying this figure can be found in https://zenodo.org/records/12783687. Figure G in S1 Text. Robustness analyses of change in Zonation priority results (difference in ranking) for the 13,195 species weighted by their IUCN status before and after prediction. We scored species without any status after prediction as 2. Species predicted as Non-Threatened were scored as 2 minus the percentage of models that predicted the status and species predicted as threatened were scored as 2 plus the percentage of models that predicted the status (scored standardized between 2 and 5). By doing so we weight more species that have been evaluated by IUCN than by our model. Biplot showing the relationship between ranks before and after prediction. Each point represents a cell. Points above x = y mean that priority rank of the given cell increases, while points below x = y mean that priority rank of the given cell decreases after addition of predicted IUCN status, color gradient indicates the species richness of cells. Secondary plot on the left: relationship between the delta ranks (rank after—rank before) of each cell and species richness (log10). The 2 red lines show the quantile regression (10% and 90%) using the rqss() (additive quantile regression smoothing) function of the R package quantreg v.5.95; points color gradient indicates the density (log10) of points from high (yellow) to low (blue). Secondary plot on the right: latitudinal gradient of species richness (log10). Points color gradient indicates the density of points from high (yellow) to low (blue). The data underlying this figure can be found in https://zenodo.org/records/12783687. Figure H in S1 Text. **Left**: relationship between the delta ranks (rank after—rank before) of each cell and species richness (log10). The 2 red lines show the quantile regression (10% and 90%) using the rqss() (additive quantile regression smoothing) function of the R package quantreg v.5.95; points color gradient indicates the density (log10) of points from high (yellow) to low (blue). **Right**: latitudinal gradient of species richness (log10). Points color gradient indicates the density of points from high (yellow) to low (blue). The data underlying this figure can be found in https://zenodo.org/records/12783687.
(DOCX)

## Acknowledgments

The authors would like to thank Miki Mori for proofreading our manuscript prior to publication.

## Author Contributions

**Conceptualization:** Nicolas Loiseau, David Mouillot, Nicolas Mouquet.

**Data curation:** Nicolas Loiseau, Laure Velez, Nicolas Casajus, Camille Albouy, Agnès Duhamet, Valentine Fleure.

**Formal analysis:** Nicolas Loiseau, Laure Velez, Nicolas Casajus, Valentine Fleure.

**Funding acquisition:** Nicolas Loiseau, David Mouillot.

**Investigation:** Nicolas Loiseau, Nicolas Mouquet.

**Methodology:** Nicolas Loiseau, Laure Velez, Raphaël Seguin, Nicolas Casajus, Camille Albouy, Valentine Fleure, Juliette Langlois, Nicolas Mouquet.

**Software:** Nicolas Loiseau, Raphaël Seguin, Nicolas Casajus.

**Validation:** Nicolas Mouquet.

**Visualization:** Nicolas Loiseau, Raphaël Seguin, Camille Coux, Nicolas Mouquet.

**Writing – original draft:** Nicolas Loiseau, Nicolas Mouquet.

**Writing – review & editing:** David Mouillot, Laure Velez, Raphaël Seguin, Nicolas Casajus, Camille Coux, Camille Albouy, Thomas Claverie, Agnès Duhamet, Valentine Fleure, Juliette Langlois, Sébastien Villéger.

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
