## [Editor Report · Decision Letter 0]

17 Jan 2024

Dear Dr Loiseau, 

Thank you for submitting your manuscript entitled "Inferring marine fish extinction risk to inform global conservation priorities" for consideration as a Research Article by PLOS Biology.

Your manuscript has now been evaluated by the PLOS Biology editorial staff, as well as by an academic editor with relevant expertise, and I'm writing to let you know that we would like to send your submission out for external peer review.

Once your full submission is complete, your paper will undergo a series of checks in preparation for peer review. After your manuscript has passed the checks it will be sent out for review. To provide the metadata for your submission, please Login to Editorial Manager (https://www.editorialmanager.com/pbiology) within two working days, i.e. by Jan 19 2024 11:59PM.

Kind regards,

Roli Roberts

Roland Roberts, PhD

Senior Editor

PLOS Biology

rroberts@plos.org

---

## [Decision Letter · Decision Letter 1]

22 Mar 2024

Dear Dr Loiseau,

Thank you for your patience while your manuscript "Inferring marine fish extinction risk to inform global conservation priorities" was peer-reviewed at PLOS Biology. It has now been evaluated by the PLOS Biology editors, an Academic Editor with relevant expertise, and by three independent reviewers. 

You'll see that reviewer #1 is very positive, and only has textual requests, suggesting a movement of material from the Discussion to the Results, wanting comment on the age (10 yrs) of the occurrence data, and asking you to mention climate change more prominently. Reviewer #2 is also positive, but wants several additional (and sensible-sounding) analyses to explore “erroneous” classification, to calculate additional performance stats and to do a sensitivity analysis; they also want clarity around how grid sizes were handled. Reviewer #3 is broadly positive, but raises a number of concerns arising from the combination of algorithms and wants more clarity around sensitivity analyses.

IMPORTANT: I discussed these comments with the Academic Editor, who agreed that we should invite a revision to address the concerns raised by the reviewers. The Academic Editor had previously mentioned two concerns; one is that you should quantify the uncertainty of each assignment (this issue also seems to be raised by reviewer #2), and the other is that you should substantially revise Figure 7: "I think it is too busy and log(10)richness should be back-transformed into something more meaningful along with maybe labelling the x-axis to explain "rank after - rank before" more intuitively, e.g. threat under-estimated vs over-estimated at different ends of the x-axis."

In light of the reviews, which you will find at the end of this email, we would like to invite you to revise the work to thoroughly address the reviewers' reports.

Given the extent of revision needed, we cannot make a decision about publication until we have seen the revised manuscript and your response to the reviewers' comments. Your revised manuscript is likely to be sent for further evaluation by all or a subset of the reviewers.

**IMPORTANT - SUBMITTING YOUR REVISION**

*Re-submission Checklist*

*Published Peer Review*

*PLOS Data Policy*

Sincerely,

Roli Roberts

Roland Roberts, PhD

Senior Editor

PLOS Biology

rroberts@plos.org

REVIEWERS' COMMENTS:

Reviewer #1:

[identifies herself as Alice Rogers]

This manuscript presents a well-considered and thoroughly explained method to predict and assign extinction risk categories to marine teleost fish, for which data are currently limited. After explaining the new methods, with care to acknowledge their strengths and short fallings, the authors are able to assign a threat category to a huge proportion of fish for which IUCN currently are not able to assess. Moreover, the authors explore the consequences of their classifications, both in terms of where hotspots of extinction might be, and how they might change with this new kind of knowledge. They present the effectiveness of current protection based on knowledge before and after this method is applied and explore the how conservation planning might change with this new classification.

I was very impressed with this paper and feel that it makes an important, novel and interesting contribution to the field. The paper is presented clearly, has excellent visuals and would be of interest to a broad range of scientists, managers and policy makers.

A few small things that I feel could use some work / clarification are as follows:

The discussion feels very long for the format of the journal and to me includes a lot of delving deeper into the results. I wonder if some of this detail could be pulled back into the results and then the discussion could more concisely consider the limitations, future directions and implications of the findings.

In the methods the fish occurrence data that drives the results comes from 2014 which is now 10 years old. I wonder if the authors could comment on the implications of this given that there could have been significant changes in species ranges or exploitation in the past 10 years?

There is only a brief mention of climate change in the paper, which feels too little given the importance of climate as a driver for extinction in the coming decades. Could the authors expand on ways in which to better capture climate vulnerability into the assessment?

I think this is a great paper and I look forward to see its publication 

Reviewer #2:

This is an interesting, novel, timely, and important work. It adds to the growing literature on modelling threat status - albeit on thousands of species from a group that received to-date less evaluation attention - marine fishes. Nevertheless, I feel like some improvements could be made to the methods and their explanations. 

First, I would be interested in some more results and discussion of the 'erroneous' classification of your models. You modelled threat for thousands of species that have such classification yet did not explore patterns in these species. These modelled vs. 'real' categories could both provide interesting insights regarding your models' performance and potential gaps, and highlight species that do have IUCN non-NS status - but is potentially erroneous. I'd like to see Fig. 3 (or better yet a confusion matrix), of your classification of non-NS species. There could be some interesting patterns there. This is especially true if your two models' consensus is that a species should be classified differently than it currently is. Look at the 22-29% of 'error' classifications - first which direction they went, do they have particular attributes. The same goes for your 'modeled' NS species.

Also, I suggest you calculate other classification performance statistics - F1, recall, precision. This will validate if your efforts to treat the class imbalance in your categories worked well. Ultimately if you predict all your non-NS species as non-threatened you'd get an accuracy of ~96%.

Your mapping of the fishes distribution is simplistic. I understand that there is a major gap in knowledge regarding their distribution however If your grid your ranges to 1*1 degrees your minimum range for a species is 1*1 degree which can be an overestimate for many species. Also, it is unclear how this relates to you 50*50 km cells for the zonation analysis. Beyond this is seems like your grid contains non-equal area cells. 

Have you conducted a sensitivity analysis on how your ranks for threatened species affects your zonation results (1 vs. 6)? Also have you considered ranking NS species differently to Non-threatened ones? (say with the middle values between the other two categories)? There are accumulating evidence that DD and also NE species are more similar to threatened species in attributes and threats.

When imputing your trait values for species you, in essence already incorporate phylogeny into your dataset indirectly. This should be acknowledged.

Minor comments

Line 17 - you give an address for the "CEFE, Univ Montpellier, CNRS, EPHE-PSL University, IRD, Montpellier, France" - but in the author list nobody is affiliated with it.

Fig. 1 - The number of species in this figure is an underestimate for some groups. For example, according to 'the reptile database' (http://www.reptile-database.org/db-info/SpeciesStat.html) there are over 12,000 species of reptiles. This will bring the percentage of NS reptile species in your figure a lot. Amphibiaweb also has many more amphibian species, Birdlife for birds, and some other sources for mammals (see also Meiri et al. 2023).

In general, I would suggest removing non-standard acronyms. Certainly 'NT' - which has a different meaning with respect to the IUCN Redlist - than used here. 

References

Meiri S., Chapple D.G, Tolley K.A., Mitchell N., Laniado T., Cox N., Bowles P., Young B.E., Caetano G.H.O., Geschke J., Böhm M., & Roll U. 2023. Done but not dusted: reflections on the first global reptile assessment and priorities for the second. Biological Conservation, 278: 109879.

Reviewer #3:

This study builds a model that aims to predict IUCN status of marine fish by utilizing random forest and artificial neural network algorithms alongside occurrence data, traits, taxonomy, and human uses. It finds that the risk of extinction for marine fish is greater than estimated by IUCN. It then compares areas of hotspots for threatened-species compared to areas currently prioritized for conservation planning. These analyses are accompanied by neat, easily-understood figures. I find this a really interesting and useful study, utilizing RF and ANN algorithms to better understand threatened status where there is no IUCN status for understudied species. 

However, I think that the low accuracy and method for combining the algorithms could be improved or at the very least accompanied with more caveats so that the reader understands the limitations of the predicted status. The low accuracy for both algorithms is slightly worrying and while I do understand that there were limitations with the number of threatened species available in the learning dataset, this does mean that the results should be presented with more caveats and uncertainties. 

Furthermore, I find the combination of machine and deep learning algorithms to be questionable - specifically with the 2nd step of the RF-ANN complementary decision tree (line 491: (ii) when one of the algorithm was not able to predict status but the second was able, the predicted status of this last one was assigned to the species). It makes sense that when both algorithms agree then that predicted status should be used, but if one algorithm simply doesn't report a status, I don't think that the assumption should be that the first algorithm is correct. It seems to me that the point of this step is a cross-check. If there is nothing else to cross-check with, shouldn't the species be assigned a No Status, as this is a form of divergence (as in option (iii))? Otherwise, this could lead to higher inaccuracy or muddy the overall dataset. 

There is mention of a couple sensitivity tests that were conducted, but these results are not expanded on much. For example, there is mention of a sensitivity test conducted for the RF-ANN algorithm combination step (line 493) where both algorithms had to predict the same result and refers to Fig. 2. However, I believe Fig. 2 displays the method reported in the paper, not the sensitivity analysis version (which I believe is Fig S1?). I think you could expand on this sensitivity test more to justify your chosen 3-option method as it seems quite a big assumption that if an algorithm doesn't predict a status, then just refer to the results of the second algorithm. There is also mention of a Zonation algorithm but the explanation is unclear and a bit too brief. Please expand on this - where do the 6 classification levels come from and how are they defined? There is again mention of a plotted sensitivity analysis (line 541) but I'm unsure which figure is associated with this - is it Fig. 7?

Overall I find this an important and interesting study that covers a gap in knowledge for threatened status of marine fishes and just needs a little more clarification on the methods and caveats for the low model accuracy.

---

## [Editor Report · Decision Letter 2]

15 Jul 2024

Dear Dr Loiseau,

Thank you for your patience while we considered your revised manuscript "Inferring marine fish extinction risk to inform global conservation priorities" for publication as a Research Article at PLOS Biology. This revised version of your manuscript has been evaluated by the PLOS Biology editors and the Academic Editor.

Based on our Academic Editor's assessment of your revision, we are likely to accept this manuscript for publication, provided you satisfactorily address the following data and other policy-related requests.

IMPORTANT - please attend to the following:

a) Please change your Title slightly to: "Inferring the extinction risk of marine fish to inform global conservation priorities"

b) There are some issues with English language/grammar throughout the manuscript, so you should re-read it carefully with this in mind; alternatively you may benefit from running it past a native English speaker, or enlist a professional editing service.

c) Because the output of this study is likely to be of broad interest to people who do not have access to the appropriate computational tools, please provide a simple spreadsheet containing the inferred IUCN status of all ~4,600 species that you analysed, for non-R-conversant readers? (there may already be one in your Github deposition, but I couldn't find it). This should be uploaded as a supplementary Table, and cited in the manuscript.

d) Many thanks for providing the underlying data and code in Github. Please could you confirm that it is sufficient to generate Figs 1, 3, 4, 5ABCD, 6AB, 7AB, and Figs S1-S8?

e) Because Github depositions can be readily changed or deleted, please make a permanent DOI’d copy (e.g. in Zenodo) and provide this URL (see next point).

f) Please cite the location of the data clearly in all relevant main and supplementary Figure legends, e.g. “The data underlying this Figure can be found in https://zenodo.org/records/XXXXXXXX

We expect to receive your revised manuscript within two weeks. 

*Published Peer Review History*

*Press*

Sincerely,

Roli Roberts

Roland Roberts, PhD

Senior Editor

rroberts@plos.org

PLOS Biology

DATA NOT SHOWN?

---

## [Editor Report · Decision Letter 3]

29 Jul 2024

Dear Dr Loiseau,

Thank you for the submission of your revised Research Article "Inferring the extinction risk of marine fish to inform global conservation priorities" for publication in PLOS Biology. On behalf of my colleagues and the Academic Editor, Andrew Tanentzap, I'm pleased to say that we can in principle accept your manuscript for publication, provided you address any remaining formatting and reporting issues. These will be detailed in an email you should receive within 2-3 business days from our colleagues in the journal operations team; no action is required from you until then. Please note that we will not be able to formally accept your manuscript and schedule it for publication until you have completed any requested changes.

IMPORTANT: I've asked my colleagues to include the following request among their own: "Many thanks for citing the location of the data in the main Figure legends; please provide similar citations in the legends for supplementary Figs S1-S8."

Sincerely,

Roli Roberts

Senior Editor

PLOS Biology

rroberts@plos.org